# Genetic Deletion or Pharmacological Inhibition of Soluble Epoxide Hydrolase Ameliorates Cardiac Ischemia/Reperfusion Injury by Attenuating NLRP3 Inflammasome Activation

**DOI:** 10.3390/ijms20143502

**Published:** 2019-07-17

**Authors:** Ahmed M. Darwesh, Hedieh Keshavarz-Bahaghighat, K. Lockhart Jamieson, John M. Seubert

**Affiliations:** 1Faculty of Pharmacy and Pharmaceutical Sciences, University of Alberta, 2026-M Katz Group Centre for Pharmacy and Health Research, 11361-97 Ave, Edmonton, AB T6G 2E1, Canada; 2Department of Pharmacology and Toxicology, Faculty of Pharmacy, Suez Canal University, 41522 Ismailia, Egypt; 3Department of Pharmacology, Faculty of Medicine and Dentistry, University of Alberta, Edmonton, AB T6G 2R7, Canada

**Keywords:** cardioprotection, ischemia-reperfusion, mitochondria, NLRP3 inflammasome, polyunsaturated fatty acids, soluble epoxide hydrolase

## Abstract

Activation of the nucleotide-binding oligomerization domain-like receptor (NLR) family pyrin domain containing 3 (NLRP3) inflammasome cascade has a role in the pathogenesis of ischemia/reperfusion (IR) injury. There is growing evidence indicating cytochrome p450 (CYP450)-derived metabolites of n-3 and n-6 polyunsaturated fatty acids (PUFAs) possess both adverse and protective effects in the heart. CYP-derived epoxy metabolites are rapidly hydrolyzed by the soluble epoxide hydrolase (sEH). The current study hypothesized that the cardioprotective effects of inhibiting sEH involves limiting activation of the NLRP3 inflammasome. Isolated hearts from young wild-type (WT) and sEH null mice were perfused in the Langendorff mode with either vehicle or the specific sEH inhibitor *t*-AUCB. Improved post-ischemic functional recovery and better mitochondrial respiration were observed in both sEH null hearts or WT hearts perfused with *t*-AUCB. Inhibition of sEH markedly attenuated the activation of the NLRP3 inflammasome complex and limited the mitochondrial localization of the fission protein dynamin-related protein-1 (Drp-1) triggered by IR injury. Cardioprotective effects stemming from the inhibition of sEH included preserved activities of both cytosolic thioredoxin (Trx)-1 and mitochondrial Trx-2 antioxidant enzymes. Together, these data demonstrate that inhibiting sEH imparts cardioprotection against IR injury via maintaining post-ischemic mitochondrial function and attenuating a detrimental innate inflammatory response.

## 1. Introduction

Ischemic heart disease is a leading cause of cardiovascular death and disability worldwide [1,2]. In patients who experience an ischemic event, early and successful restoration of blood flow to the ischemic myocardium, a process known as reperfusion, is critical to maintain viable myocardial tissue, limit infarct size and reduce acute mortality rates. However, reperfusion paradoxically can induce and exacerbate tissue injury resulting in increased incidence of chronic heart failure. Several studies demonstrated that up to 50% of the final infarct size could be attributed to the ischemia/reperfusion (IR) insult [3,4]. While the mechanisms underlying IR injury are complex, activation of an inflammatory response associated with excessive mitochondrial damage contributes to deteriorating heart function [5]. 

Experimental evidence has demonstrated that during reperfusion a surge of reactive oxygen species (ROS) is rapidly generated from damaged mitochondria. This ROS burst triggers a series of inflammatory reactions, which induce the formation and activation of inflammasomes aggravating myocardial injury [6,7,8]. Inflammasomes are large cytosolic inflammatory protein scaffolds assembled in response to cellular danger signals in order to activate several innate immune defenses [9,10]. The most widely characterized inflammasome platform in the heart that becomes activated in response to aseptic stimuli, such as myocardial IR injury, is the nucleotide-binding oligomerization domain-like receptor (NLR) family pyrin domain containing 3 (NLRP3) [11,12]. In the setting of a myocardial IR insult, activation of the NLRP3 inflammasome spreads an inflammatory surge to the rest of the myocardium triggering further damage by promoting the autocatalytic activation of pro-caspase-1. Active caspase-1 subsequently cleaves inactive pro-interleukin-1beta (pro-IL-1β) into to the mature pro-inflammatory cytokine IL-1β triggering pyroptosis or caspase-1 mediated cell death [7,13,14,15]. Accordingly, limiting mitochondrial damage and blunting the activation of a NLRP3 inflammasome cascade is a promising therapeutic strategy to promote recovery and alleviate adverse cardiac injury following IR insult. 

Soluble epoxide hydrolase (sEH) catalyzes the hydrolysis of lipid epoxides to their corresponding diol derivatives by the addition of water [16]. sEH is highly expressed in the mammalian heart tissue and has a pivotal role in metabolizing cytochrome P450 (CYP450)-derived epoxy metabolites of both n-3 and n-6 polyunsaturated fatty acids (PUFAs) to their corresponding diol derivatives [17]. For example, sEH rapidly degrades the cardioprotective lipids epoxydocosapentaenoic acids (EDPs) and epoxyeicosatrienoic acids (EETs) into their less bioactive corresponding diol products [18,19]. In contrast, the CYP-derived epoxy metabolites of linoleic acid (LA), epoxyoctadecenoic acids (EpOMEs), are rapidly converted by sEH into their corresponding bioactive diols, dihydroxyoctadecenoic acid (DiHOME) [20,21], which have been shown to have cardiotoxic effects [22,23,24,25]. Altogether, the detrimental outcomes associated with the activation of sEH in response to cardiovascular insult could be attributed to the excessive degradation of protective epoxylipids (i.e., EDPs and EETs) and enhanced production of toxic diol-metabolites (i.e., DiHOMEs).

Inhibition of sEH has emerged as an intriguing approach to limit cardiac damage in different cardiovascular settings [26,27]. Despite all the promising findings associated with the inhibition of sEH in the heart, the effect of sEH inhibition on mitochondrial degeneration and the associated NLRP3 inflammasome activation in the setting of IR injury has not been investigated. Results from the current study builds upon our previous findings [28,29] and demonstrates that deletion of the gene encoding sEH (*Ephx2*) or pharmacological inhibition of sEH enzyme could attenuate myocardial IR injury through maintaining mitochondrial function and consequently limiting NLRP3 inflammasome activation. To the best of our knowledge, the current study also provides the first evidence that the cardioprotective effects associated with sEH inhibition against IR injury is sex-independent in young subjects. 

## 2. Results

### 2.1. Deletion or Inhibition of sEH Improves Post-Ischemic Functional Recovery in Both Males and Females

Although accumulating literature suggests that sEH enzyme is a good target to ameliorate IR injury [24,26,30], the differential response of both males and females to sEH inhibition has not been investigated. Notably, preischemic cardiac parameters were similar between males and females in all young treatment groups. sEH null hearts or WT hearts perfused with the sEH inhibitor *trans*-4-[4-(3-adamantan-1-yl-ureido)-cyclohexyloxy]-benzoic acid (*t*-AUCB) and subjected to IR showed significantly improved post-ischemic recovery of left ventricular developed pressure (LVDP) compared to the WT IR control group (Figure 1A). Importantly, both male and female hearts respond similarly to the genetic deletion of *Ephx2* or pharmacological inhibition of sEH as there were no significant differences between both genders in terms of post-ischemic functional recovery (Figure 1A). Moreover, no significant differences were observed in the heart rate at the end of reperfusion between all the study groups (Figure 1B). Consistent with improved post-ischemic functional recovery, both male and female sEH null hearts or WT hearts perfused with *t*-AUCB demonstrated better rates of contraction (dP/dt max) and relaxation (dP/dt min) in comparison to the corresponding IR control mice (Figure 1C,D). Interestingly, the coronary flow rates did not significantly differ between pre- and post-ischemic perfused heart in any of the treatment groups in our model indicating the cardioprotective effect was not attributable to alterations in hemodynamics in the perfused heart model (Figure 1E). Together, these data suggest that the genetic deletion of *Ephx2* or pharmacological inhibition of sEH similarly improve post-ischemic functional recovery in both males and females and as such data from both sexes were combined in the rest of the experiments.

### 2.2. Deletion or Inhibition of sEH Limits Post-Ischemic Mitochondrial Injury

Mitochondria serve as the important arbiters of cardiomyocyte life and death [31]. Impaired mitochondrial function, associated with excessive ROS production, secondary to IR injury leads to a vicious cycle of continued injury and reduced cardiac contractile function [32,33]. Accordingly, we investigated the effect of the genetic deletion of *Ephx2* or pharmacological inhibition (*t*-AUCB) of sEH on the mitochondrial respiration in fibers separated from hearts subjected to IR injury. Notably, basal respiration rates did not differ significantly between all groups (Table 1), however, respiratory control ratio (RCR), a marker of mitochondrial efficiency, was significantly reduced in post-ischemic WT vehicle control hearts compared to both the WT and sEH null aerobic controls (Figure 2). However, genetic deletion of *Ephx2* or pharmacological inhibition of sEH preserved post-ischemic adenosine diphosphate (ADP)-stimulated respiration and RCR values, suggesting better mitochondrial function in post-ischemic hearts (Table 1). 

Several studies demonstrated that proteins regulating mitochondrial dynamics, such as dynamin-related protein 1 (Drp-1), play a role in the cascade of myocardial IR injury [34,35]. In response to IR injury, Drp-1 translocates from the cytosol to the mitochondria resulting in uncontrolled and exaggerated fission inducing myocardial cell death [36]. Consistent with this notion, we observed a significant increase in mitochondrial expression of Drp-1 in hearts from WT IR mice (Figure 3A). This finding is consistent with the decreased mitochondrial respiration/function observed in post-ischemic vehicle WT IR hearts. Genetic deletion of *Ephx2* or pharmacological inhibition of sEH limited the post-ischemic mitochondrial localization of Drp-1 (Figure 3A), supporting the notion of reduced mitochondrial injury. 

Excessive mitochondrial damage, in response to IR injury, often results in significant elevations in cellular ROS levels. Thioredoxins (Trxs) are important antioxidant proteins that play a cytoprotective role against various oxidative stresses in a variety of systems. Trxs are important for maintaining the reducing environment in the cell, protecting against oxidative stress and thus limiting cardiomyocyte cell death [37,38]. In mammalian cells, there are two major isoforms of Trxs, cytosolic Trx-1 and mitochondrial Trx-2 [37,39,40]. In the current study, we observed a marked reduction in both cytosolic Trx-1 and mitochondrial Trx-2 antioxidant activities in WT hearts subjected to IR injury, suggesting increased ROS levels (Figure 3B,C). Moreover, the cardiac levels of malondialdehyde (MDA), the main end product of lipid peroxidation and a key marker of oxidative stress [41,42], were significantly elevated following IR injury (Figure 3D). However, genetic deletion of *Ephx2* or pharmacological inhibition of sEH preserved both Trx-1 and Trx-2 catalytic activities as well as significantly blunted the accumulation of MDA in post-ischemic hearts (Figure 3B–D), suggesting less ROS production. 

### 2.3. Deletion or Inhibition of sEH Abrogates the Assembly and Activation of NLRP3 Inflammasome Secondary to IR Injury

Immunoblotting confirmed the complete ablation of the gene encoding sEH (*Ephx2*) in sEH null animals (Figure 4A). Furthermore, there was no marked change in the expression of sEH protein in WT mice subjected to IR injury compared to their counterparts under aerobic conditions (Figure 4A). A direct correlation between mitochondrial dysfunction, excessive ROS production and the activation of NLRP3 inflammasome cascade has been well-established in several reports [8,43]. To determine the effect of the genetic deletion of *Ephx2* or pharmacological inhibition of sEH on the activation of the NLRP3 inflammasome, we assessed the expression levels of NLRP3 and IL-1β proteins as well as caspase-1 activity. Immunoblot analyses showed that IR injury markedly up-regulated NLRP3 protein expression (Figure 4B), which correlated with increased catalytic activity of caspase-1 in WT hearts (Figure 4C). Moreover, enzyme-linked immunosorbent assay (ELISA) results revealed that active cytokine IL-1β was increased following IR injury in WT hearts (Figure 4D). Genetic ablation of *Ephx2* or perfusing with *t*-AUCB markedly abrogated the IR-induced upregulation of NLRP3 and IL-1 β protein expression and prevented the increase of caspase-1 activity (Figure 4B–D). Altogether, these data suggest inhibition of sEH limits NLRP3 inflammasome activation correlated with reduced cardiac IR injury.

### 2.4. Deletion or Inhibition of sEH Blunts the Activation and Mitochondrial Translocation of Thioredoxin-Interacting Protein (Txnip) Secondary to IR Injury

Thioredoxin-interacting protein (Txnip) is an upstream trigger in the NLRP3 inflammasome cascade that binds these two proteins together, which is essential for downstream activation. Furthermore, Txnip acts as a pro-oxidant protein by preferentially binding to and inhibiting the antioxidant activities of both Trx-1 and -2. Thus, Txnip serves as a central protein linking oxidative stress to NLRP3 inflammasome formation [44,45,46]. In the current study, post-ischemic WT hearts had a significant increase of both cytosolic and mitochondrial Txnip protein expression consistent with increased inflammasome formation and mitochondrial dysfunction (Figure 5A,B). Interestingly, the accumulation of Txnip correlates with the significant reduction in the antioxidant activities of both Trx-1 and -2 in WT hearts following IR injury (Figure 3B,C). In contrast, hearts perfused with *t*-AUCB or isolated from sEH null mice demonstrated significantly lower post-ischemic localization of Txnip in both cytosolic and mitochondrial fractions (Figure 5A,B). 

### 2.5. Deletion or Inhibition of sEH Abrogates the IR-Induced Upregulation of the Mitochondrial Protein Mitofusin-2

Mitofusins (Mfn)-1 and -2 are integral proteins that localize to the outer mitochondrial membrane and have a role in dynamic events regulating mitochondrial quality [47]. In the mammalian heart, Mfn-1 expression is higher than Mfn-2 and is primarily responsible for mitochondrial fusion [48]. However, emerging evidence indicates non-canonical roles for Mfn-2 within specific cell types include activation of the NLRP3 inflammasome and cardiomyocyte cell death [49,50]. In the current study, we observed no significant differences in mitochondrial expression of Mfn-1 among the different study groups (Figure 6A). However, there were significant increases in mitochondrial expression of Mfn-2 in post-ischemic WT hearts, which were attenuated when *Ephx2* was genetically deleted (Figure 6B). Although pharmacological inhibition of sEH showed a trend to limit mitochondrial expression of Mfn-2, it did not reach statistical significance compared to WT IR hearts (Figure 6B). Overall, these data correlate with deteriorated mitochondrial function and activated NLRP3 inflammasome cascade observed in post-ischemic WT hearts and are also consistent with the reduced cardiac injury in sEH null or pharmacologically treated mice. 

## 3. Discussion

In the current study, we demonstrate that genetic deletion of *Ephx2* or pharmacological inhibition of sEH enzyme with *t*-AUCB limits mitochondrial damage, abrogates activation of the NLRP3 inflammasome cascade, improves post-ischemic functional recovery and thus imparts cardioprotection in the setting of IR injury. The cardioprotective effects associated with sEH inhibition were independent of sex or hemodynamic changes in young mice. Together, the current data suggest a novel cardioprotective mechanism for inhibition of sEH that limits an innate inflammatory response.

The gene encoding mammalian sEH (*Ephx2*) has been identified in numerous species such as mouse [51], rat [52] and human [53], as well as bacteria [54,55] and plants [56,57]. Importantly, sEH catalytic activity supports both advantageous and deleterious reactions in its role to metabolize endogenous or exogenous epoxides. Under normal physiological conditions, endothelial sEH plays a pivotal role in the metabolism of epoxylipids into diol metabolites, a function that modulates the biological effects of the lipids within the cardiovascular system [58]. Moreover, in healthy individuals, there is likely an equilibrium between the anti-inflammatory metabolites generated by CYP450 and the sEH-derived pro-inflammatory ones [59]. However, under many pathophysiological states, such as diabetes, obesity, IR injury and aging, there is a shift in equilibrium favoring more sEH-dependent inflammatory pathways [60]. Therefore, blocking sEH activity has become a promising therapeutic approach to limit adverse inflammatory responses and the associated injury.

Inhibition of sEH has emerged over the last few years as an attractive therapeutic approach for the treatment and prevention of several cardiovascular disorders [24,26,27,30,61,62,63,64,65]. In the mammalian heart, there is a mounting evidence that CYP-derived epoxylipids EETs and EDPs mediate many beneficial effects by maintaining mitochondrial quality and reducing adverse inflammatory reactions [24,30,65,66,67,68,69]. Importantly, the rapid hydrolysis of the bioactive oxylipins by sEH limits their beneficial cardiovascular effects while increasing adverse effects [19]. Previous data has demonstrated deficiency of sEH is associated with improved post-ischemic functional recovery with smaller infarct size [26,70]. These beneficial effects were attributed in part to the stabilization of protective epoxy-metabolites EETs and EDPs [29,71,72]. In contrast, CYP-derived epoxylipids EpOMEs are metabolized by sEH to DiHOMEs, which possess more cytotoxic effects [22,23,24,73,74]. Early evidence has demonstrated that the cardiac levels of DiHOMEs are increased in models of myocardial IR injury [75]. Recent data indicates both 12,13-EpOME and 12,13-DiHOME diminish post-ischemic cardiac functional recovery, however, inhibition of sEH prevented the detrimental effect of 12,13-EpOME suggesting 12,13-DiHOME was the active metabolite [25]. Moreover, Edin et al. reported that increased post-ischemic functional recovery in sEH-deficient mice was associated with higher concentrations of EETs and lower concentrations of LA diols [76]. The exact cardiotoxic mechanisms of DiHOMEs are unknown but are most likely attributable to several different effects. For example, accumulation of DiHOMEs in the heart is associated with impaired mitochondrial function, uncoupled oxidative phosphorylation and altered ion channel kinetics resulting in extensive cardiac injury [22,70,76,77,78]. Taken together, these reports indicate that inhibition of sEH could serve as a dual cardioprotective strategy to preserve the levels of cardioprotective epoxylipids while simultaneously decreasing the production of the cardiotoxic LA diols, ultimately protecting cardiac mitochondria and preserving cardiac function (Figure 7). 

Excessive activation of the innate immune system and the associated inflammatory response plays a pivotal role in aggravating myocardial IR injury [79,80]. Recently, the innate NLRP3 inflammasome cascade has been found to be a major contributor to the pathology [81,82]. Briefly, the death of cardiomyocytes resulting from acute ischemic conditions or reperfusion injury causes the release of cellular debris and contents, referred to as damage-associated molecular patterns (DAMPs) [83,84]. Binding of DAMPs to the pattern recognition NOD-like receptors (NLR) on cardiac fibroblasts, infiltrating leucocytes and cardiomyocytes will activate the oligomerization and formation of NLRP3 inflammasomes [82,85]. Moreover, several studies demonstrated that the activation of the NLRP3 protein, the main component of NLRP3 inflammasome, after IR injury is attributed to the cross-talk between NLRP3 inflammasome and mitochondria, whereby NLRP3 senses ROS produced by dysfunctional mitochondria [8,15]. Once aggregated, NLRP3 inflammasome mediates the cleavage and activation of caspase-1. Active caspase-1 then induces the conversion of pro-IL-1β to mature IL-1β [7,13,14,82,85,86]. IL-1β triggers the release of other cytokines and chemokines, which recruit and activate inflammatory cells such as neutrophils and monocytes driving a severe inflammatory process aggravating cellular injury [9]. Accumulating literature demonstrates the main role of IL-1β involves initiating the inflammatory cascade, however evidence suggests it may have a direct detrimental effect on the myocardium. For instance, it has been reported IL-1β acts as a cardio-depressant cytokine where single or multiple injections of IL-1β causes systolic dysfunction and reduces LV contractility reserve in healthy mice and human subjects in the absence of ischemia [87,88,89,90]. Furthermore, IL-1β induces direct negative inotropic effects in isolated perfused rat hearts [91]. Moreover, in vitro experiments have demonstrated IL-1β stimulation activates apoptotic pathways in neonatal rat cardiomyocytes [92]. Our current understanding indicates IL-1β triggers cardiac damage in IR injury through either recruitment of a pro-inflammatory cells, such as leukocytes, causing adverse response or direct action on cardiac cells [12,79,93,94]. 

Deletion of the NLRP3 gene or pharmacological inhibition of inflammasome formation post-MI is associated with cardiovascular protection resulting in smaller infarct size and better functional recovery [7,13,14,44,82,86,95,96,97,98]. Increasing evidence from both human and animal studies demonstrate some CYP-derived epoxylipids, such as EDPs and EETs, possess anti-inflammatory properties. Therefore, therapeutic approaches to elevate the levels of these lipid mediators, as in the case of sEH inhibition, are useful for treating different inflammatory disorders such as diabetes, atherosclerosis, and arthritis [67,99,100,101,102]. Zhou et al., demonstrated inhibition of sEH can attenuate lipopolysaccharide (LPS)-induced acute lung injury and improve survival in mice by suppressing the activation of NLRP3 inflammasome and the expression of its downstream effector IL-1β [103]. Recently, we demonstrated EDPs can suppress NLRP3 inflammasome activation thereby inhibiting production of proinflammatory cytokines including IL-1β following IR injury [104]. The current study provides supporting evidence whereby the cardioprotective effects following pharmacological inhibition of sEH or genetic deletion of *Ephx2* are attributable to a reduced innate immune response.

Mitochondria are important organelles required for healthy cardiac function but are susceptible to significant injury from ischemia and reperfusion. IR-induced injury causes damage to the electron transport chain (ETC) mainly during the ischemic period [105]. As such, re-introduction of oxygen to the ischemic myocardium that contains injured mitochondria leads to mitochondrial-driven injury with excessive production of ROS and accumulation of calcium, which contributes to contractile dysfunction and cell death [106]. Recent evidence suggests mitochondria play a key role in regulating NLRP3 inflammasome activation and the subsequent responses [7]. The current study demonstrates cardioprotective effects associated with sEH inhibition involve reduced mitochondrial injury, which is attributed to an altered balance between CYP-derived epoxylipids, EETs and EDPs, with sEH cardiotoxic diols, DiHOMEs. 

Proteins regulating mitochondrial dynamics are involved in the pathogenesis of myocardial IR injury. Under normal conditions, Drp-1 is localized in the cytosol in an inactive phosphorylated form, which becomes dephosphorylated following a stress event and translocates to the outer mitochondrial membrane initiating mitochondrial fission, reducing the number of functional mitochondria and accelerating myocardial injury [34,107]. Several studies show genetic deletion or pharmacological inhibition of Drp-1 protects cardiomyocytes against IR injury and improves cardiac contractile function [35,36]. Data generated from the current study support these reports where increased mitochondrial Drp-1 in the IR group correlates with reduced mitochondrial respiration and impaired cardiac functional recovery. Both genetic deletion of *Ephx2* and pharmacological inhibition of sEH enzyme significantly inhibited the translocation of Drp-1 to the mitochondria, correlating with improved cardiac functional recovery. These results support our previous findings demonstrating post-MI mitochondrial ultrastructure and function in young WT mice displayed a complete loss of cellular organization, cristae and function that was maintained in the young sEH null mice [29,72].

Evidence demonstrates Txnip, a pro-oxidant and a well-known activator of NLRP3 inflammasome, has a negative role in the pathogenesis of IR injury [46,108]. Under normal physiological conditions, Txnip is confined primarily to the nucleus, however, following IR stress and excessive ROS production, it translocates to the cytosol inhibiting the antioxidant activity of Trx-1 and activates the oligomerization of NLRP3 inflammasome. Active NLRP3 inflammasome in turn shuttles and translocates Txnip to the mitochondria inhibiting the main mitochondrial antioxidant Trx-2 initiating the process of cardiomyocyte cell death [45,109,110]. Similarly, the current study demonstrates IR injury is associated with increased cytosolic and mitochondrial Txnip, reduced Trx-1 and Trx-2 activities as well as accumulated MDA levels, the main lipid peroxidation end product suggesting increased ROS production [41,111]. Yoshioka et al. demonstrated that genetic deletion of Txnip protects the myocardium from IR injury [112]. Consistent with this notion, our results indicate deficiency of sEH inhibited the translocation of Txnip to the cytosol and mitochondria under IR conditions, limited the loss of the antioxidant activities of both Trx-1 and -2 proteins, ameliorated the accumulation of MDA and correlated with improved post-ischemic functional recovery. 

Emerging evidence for the non-canonical roles of Mfn-2 in the activation of different innate immune components and pathogenesis of IR injury indicates that mitochondrial dysfunction is associated with increased Mfn-2 expression, which accelerates cardiomyocyte death [113,114]. The exact mechanisms remain unknown, Mfn-2 mediated tethering of mitochondria and endoplasmic reticulum facilitate the transfer of calcium to the mitochondria. Under ischemic conditions, this phenomenon accumulates calcium in the mitochondria accelerating degradation [115,116]. Indeed, acute ablation of cardiac Mfn-2 reduces mitochondrial calcium overload, rendering the heart resistant to acute infarction following IR [117]. The current study demonstrated myocardial IR injury was associated with increased mitochondrial Mfn-2 protein expression but was prevented by inhibition of sEH. Altogether, we believe that deficiency of sEH limits the mitochondrial damage in response to injury; however, the connection between Mfn-2 and the protective mechanism(s) needs further investigation. 

In conclusion, our results demonstrate that inhibition of sEH protects against myocardial IR injury by preserving mitochondrial function and inhibiting NLRP3 inflammasome activation. Although the exact molecular mechanisms remain unknown, we propose inhibiting sEH results in altered cardiac levels of bioactive epoxylipids EDPs and EETs together with reduced DiHOME levels, which collectively maintain an optimally functioning mitochondrial pool, inhibit a detrimental innate inflammasome response and thereby promote cell survival. It is recognized the current study was limited by not directly assessing changes in the levels of epoxylipids (EETs, EDPs, EpOMEs) as well as the diol metabolites (DiHOMEs) in response to IR injury. However, it has been shown that cardiac ischemic insults decrease levels of the cardioprotective epoxylipids EETs and EDPs and cardiotoxic DiHOMEs accumulate aggravating the injury [75,118]. Therefore, future characterization of how IR injury alters the levels of these metabolites will provide more insight into the complex responses and pathobiology.

## 4. Materials and Methods

### 4.1. Animals

Mice with targeted disruption of *EPHX2* (sEH null) and wild-type (WT) littermates on a C57/Bl6 background were maintained in a colony at the University of Alberta [24] and used in the current study. All studies were carried out using 2–3 month-old male and female mice weighing 25–30 g. Mice were fed on a standard rodent chow diet ad libitum (fat 11.3%, fiber 4.6%, protein 21% (*w*/*w*)), more specifically linoleic acid (2.12%), linolenic acid (0.27%), arachidonic acid (0.01%), omega-3 fatty acid (0.45%), total SFA (0.78%) and total MSFA (0.96%) (PicoLab^®^ Rodent Diet 20 Cat. No 5053, LabDiets, Inc., St. Louis, MO, USA) and housed under conditions of constant temperature and humidity with a 12:12-h light–dark cycle. All animal experimental protocols were approved by the University of Alberta Health Sciences Welfare Committee (University of Alberta Animal Welfare, ACUC, study ID#AUP330, Renewal June, 2019) and conducted according to strict guidelines provided by the Guide to the Care and Use of Experimental Animals (Vol. 1, 2nd ed., 1993, from the Canadian Council on Animal Care). 

### 4.2. Isolated Heart Perfusion

Soluble epoxide hydrolase null (sEH^–/–^) and wild-type (WT) mice of both sexes (equal ratios) were anesthetized by an intraperitoneal injection of sodium pentobarbital (Euthanyl, 100 mg/kg). Following complete non-responsiveness to external stimulation, hearts were quickly excised and perfused in the Langendorff mode with Krebs–Henseleit buffer containing (in mM) 120 NaCl, 25 NaHCO_3_, 10 Dextrose, 1.75 CaCl_2_, 1.2 MgSO_4_, 1.2 KH2PO_4_, 4.7 KCL, 2 Sodium Pyruvate (pH 7.4) and bubbled with 95% O_2_ and 5% CO_2_ at 37 °C [24,30,104,119]. The left atrium was then excised, and a water-filled balloon made of saran plastic wrap was inserted into the left ventricle through the mitral valve. The balloon was connected to a pressure transducer for continuous measurement of LVDP and heart rate (HR). Hearts with persistent arrhythmias or LVDP less than 80 cm H_2_O were excluded from the experiment. Mouse hearts were perfused in the retrograde mode at a constant flow rate for 40 min of baseline (stabilization) and then subjected to 30 min of global no flow ischemia followed by 40 min of reperfusion. In a group of WT mice, the specific sEH inhibitor *t*-AUCB (0.1 µM) (Cayman Chemicals, Ann Arbor, MI, USA) [28] was added 20 min before ischemia and was present in the heart until the end of the reperfusion period. The percentage of left ventricular developed pressure (%LVDP) at 40 min of reperfusion (R40), as compared to baseline LVDP, was taken as a marker for recovery of contractile function. At the end of reperfusion, hearts were immediately flash frozen in liquid nitrogen and stored below −80 °C. Contractile and hemodynamic parameters were acquired and analyzed using ADI software from (Holliston, MA, USA). Collection of the heart effluent was done during both pre- and post-ischemic protocols to determine coronary flow (CF) rates. 

### 4.3. Immunoblotting 

Frozen mouse hearts were ground, homogenized and then fractionated into mitochondrial and cytosolic fractions as previously described [30,69]. Briefly, frozen cardiac tissues were ground with mortar and pestle on dry ice and then homogenized in ice-cold homogenization buffer (20 mmol/L Tris–HCL, 50 mmol/L NaCl, 50 mmol/L NaF, 5 mmol/L sodium pyrophosphate, 1 mmol/L EDTA, and 250 mmol/L sucrose added on the day of the experiment, pH 7.0). Samples were first centrifuged at 800× *g* for 10 min at 4 °C to separate the cellular debris. The collected supernatant was then centrifuged at 10,000× *g* for 20 min. The pellet was resuspended in homogenization buffer to obtain a mitochondrial-enriched fraction. The supernatant was ultra-centrifuged at 105,000× *g* for 60 min and the subsequent supernatant was used as the cytosolic fraction. Protein concentrations in both cytosolic and mitochondrial fractions were measured by the Bradford assay. Western blotting was done as previously described [72,104]. Protein (30–50 μg) was resolved by electrophoresis on (10–15%) SDS-polyacrylamide gels and transferred onto polyvinylidene difluoride (PVDF) membranes (BioRad Laboratories, Hercules, CA, USA). Immunoblots were probed with antibodies to Drp-1 (Cat#: 5391), glyceraldehyde 3-phosphate dehydrogenase (GAPDH) (Cat#: 51745), Hsp60 (Cat#: 4870) (1:1000, Cell Signaling Technology, Inc., Danvers, MA, USA), NLRP3 protein (1:500) (Cat#: ab214185), Mfn-1 (ab104274), Mfn-2 (ab50838) (1:1000, Abcam, Burlingame, CA, USA), sEH (Cat#: E-AB-60489, 1:250, Elabscience Biotechnology Co., Wuhan, China), and Txnip (Cat#: K0205-3, 1:500, MBL International Co., Woburn, MA, USA). After washing, membranes were incubated with the corresponding secondary antibodies (1:5000). The blots were visualized with ECL reagent. Relative band intensities were expressed as fold of the control assessed using ImageJ software (Version 1.47v, NIH, USA). 

### 4.4. Enzyme-Linked Immunosorbent Assay

Enzyme-linked immunosorbent assay (ELISA) was used to quantify the cardiac cytosolic levels of the cytokine IL-1β where mouse IL-1β ELISA kit (ab100705, Abcam) was used according to the manufacturer’s recommendations. Briefly, cytosolic samples were pipetted into a 96-well plate where IL-1β present in a sample became attached to the wells by the immobilized antibody specific for mouse IL-1β that is coated on the wells. The wells were then washed, and biotinylated anti-mouse IL-1β antibody was added. Horseradish peroxidase (HRP) conjugated streptavidin was added to the wells after washing away unbound biotinylated antibody. A 3,3′,5,5′-Tetramethylbenzidine (TMB) substrate solution was then added to the wells. Afterwards, the stop solution was pipetted into the wells and the intensity of the color was measured at 450 nm. IL-1β concentration in the different samples was calculated by using a linear standard curve created with different concentrations of the standard IL-1β.

### 4.5. Measurement of MDA Levels

The level of MDA was assessed in the cardiac tissue using a lipid peroxidation (MDA) colorimetric assay kit (Abcam, Burlingame, CA, USA) according to manufacturer’s instructions [42]. In this assay, free MDA present in the sample reacts with thiobarbituric acid (TBA) and generates a MDA-TBA adduct which was quantified colorimetrically at wavelength 532 nm. MDA levels were expressed as nmole MDA per mg protein.

### 4.6. Mitochondrial Respiration

Clark electrode connected to an Oxygraph Plus recorder (Hansatech Instruments Ltd., Norfolk, England) was used to measure mitochondrial oxygen consumption in permeabilized cardiac fibers. Fresh cardiac fibers were isolated from the left ventricles of the perfused hearts at the end of reperfusion as previously described [72,120]. Briefly, heart tissues were dissected under a dissecting microscope in ice-cold isolation buffer (2.77 mM Ca KEGTA, 7.23 mM K_2_EGTA, 20 mM imidazole, 20 mM taurine, 49 mM K-MES, 3 mM K_2_HPO_4_, 9.5 mM MgCl_2_, 5.7 mM ATP,1 µM leupeptin, 15 mM phosphocreatine). A 3–5 mm strip of the anterior left ventricle was isolated and the remaining fats and vessels were removed. Afterwards, myocardial strips were disassembled into bundles containing 6–8 fibers each, 1 mm wide and 3–4 mm long. Fresh fibers were then permeabilized in isolation buffer containing 100 µg/mL saponin, washed three times for 5 min in ice-cold respiration buffer and immediately added to the respiration chamber containing 1.8 mL respiration buffer. The rate of oxygen consumption was measured at 30 °C before and after addition of 0.5 mM ADP in the presence of 5 mM malate and 10 mM glutamate as respiratory substrates to initiate basal respiration. RCR was calculated as the ratio between basal and ADP-stimulated respiration rates to estimate mitochondrial respiration efficiency. 

### 4.7. Enzymatic Assays

Cleavage of the caspase-1 specific fluorogenic substrate Ac-YVAD-AMC (Cat #: ALX-260-024-M005, Enzo life Sciences, Farmingdale, NY, USA) was used to assess functional caspase-1 activity in cytosolic fractions of the heart homogenates [121]. The assay quantitated the fluorescence intensity of the cleaved 7-Amino-4-methylcoumarin (AMC) using a fluorometer (at excitation 380 nm, and emission 460 nm wavelengths). The activity was calculated by using a linear standard curve created with AMC and normalized to the sample protein concentration.

The insulin disulfide reduction assay was conducted to measure thioredoxin (Trx) activity as previously described [122,123]. In this assay, Trx is first reduced by TrxR enzyme and then is used to reduce insulin disulfides. Briefly, equal amounts of mitochondrial or cytosolic protein (30 µg) were preincubated with 2 μL of dithiothreitol (DTT) activation buffer (100 mM HEPES (pH 7.6), 2 mM EDTA, 1 mg/mL bovine serum albumin (BSA), 2 mM DTT) at 37 °C for 15 min to reduce and activate endogenous Trx. Afterwards, 20 μL of reaction mixture containing 100 mM HEPES pH 7.6, 2 mM EDTA, 0.2 mM NADPH, and 140 μM insulin were added. The reaction was then started by the addition of 0.5 U mammalian TrxR (Cayman Chemicals, Ann Arbor, MI, USA) or an equal volume of water for negative controls. The samples were incubated at 37 °C for 30 min. The reaction was stopped by the addition of 125 μL of stop solution containing 10 M guanidine hydrochloride and 1.7 mM (5,5-dithio-bis-(2-nitrobenzoic acid) (DTNB) in 0.2 M Tris–HCl (pH 8.0). Reduction of DTNB to 5-thio-2-nitrobenzoic acid (TNB) was detected by optical density at 412 nm. Changes in the absorbance in the absence of TrxR were subtracted from those in the presence of the reductase. The Trx activity was expressed as µmol/min/mg protein.

### 4.8. Statistics

Values are expressed as mean ± standard error of mean (SEM). Statistical significance was determined by one-way analysis of variance (ANOVA) with a Tukey post hoc test to assess differences between groups; *p* < 0.05 was considered statistically significant.

## Figures and Tables

**Figure 1 ijms-20-03502-f001:**
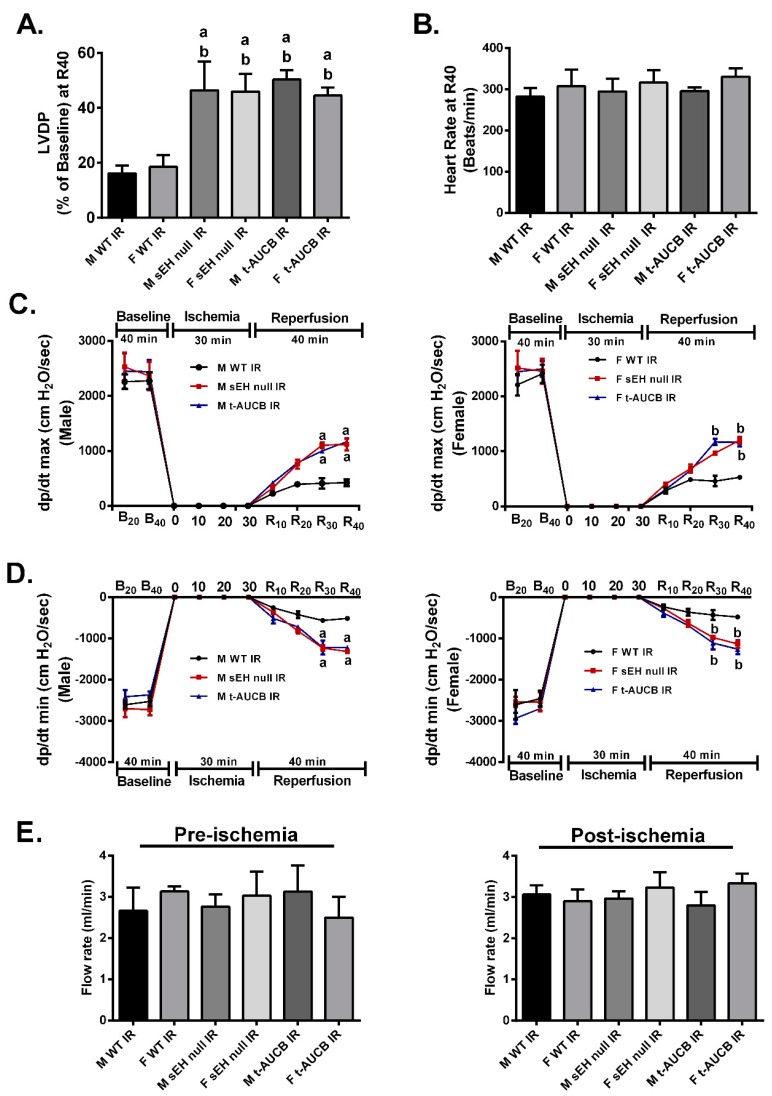
Genetic deletion of *Ephx2* or pharmacological inhibition (*t*-AUCB) of soluble epoxide hydrolase (sEH) improved post-ischemic contractile parameters in both males and females. (**A**) Left ventricular developed pressure (LVDP) recovery at 40 min of reperfusion as a percentage of baseline, (**B**) heart rate assessed as beats per minute (BPM) at the end of reperfusion (R40), (**C**) rate of contraction (dP/dt max), (**D**) rate of relaxation (dP/dt min) in both male and female hearts at the baseline before (B20) and after (B40) drug treatment, at ischemia, and at 10, 20, 30 and 40 min reperfusion (R10, R20, R30, and R40), and (**E**) coronary flow rates from perfused hearts both pre- and post-ischemia. Values represent mean ± standard error of mean (SEM); ^a^
*p* < 0.05 vs. M WT IR, ^b^
*p* < 0.05 vs. F WT IR (*n* = 4–7 per group). F; Female, LVDP; Left ventricular developed pressure, M; Male.

**Figure 2 ijms-20-03502-f002:**
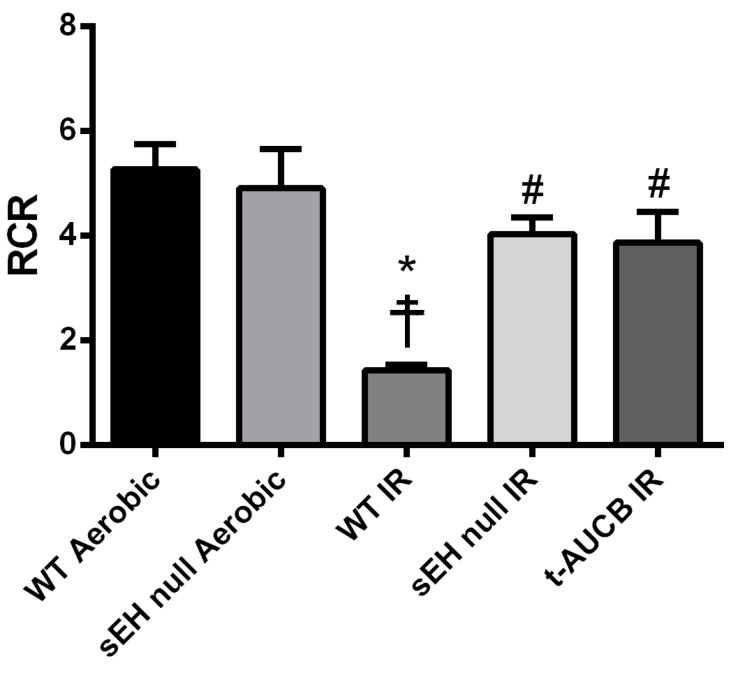
Deficiency of sEH preserved mitochondrial respiratory function following IR injury. Histogram demonstrating changes in respiratory control ratio (RCR) values in both wild-type (WT) and sEH null mice under aerobic conditions or following 30 min ischemia and 40 min reperfusion. Values represent mean ± SEM, * *p* < 0.05 vs. WT Aerobic CT, ☨ *p* < 0.05 vs. sEH null Aerobic, # *p* < 0.05 vs. WT IR (*n* = 4–8 per group).

**Figure 3 ijms-20-03502-f003:**
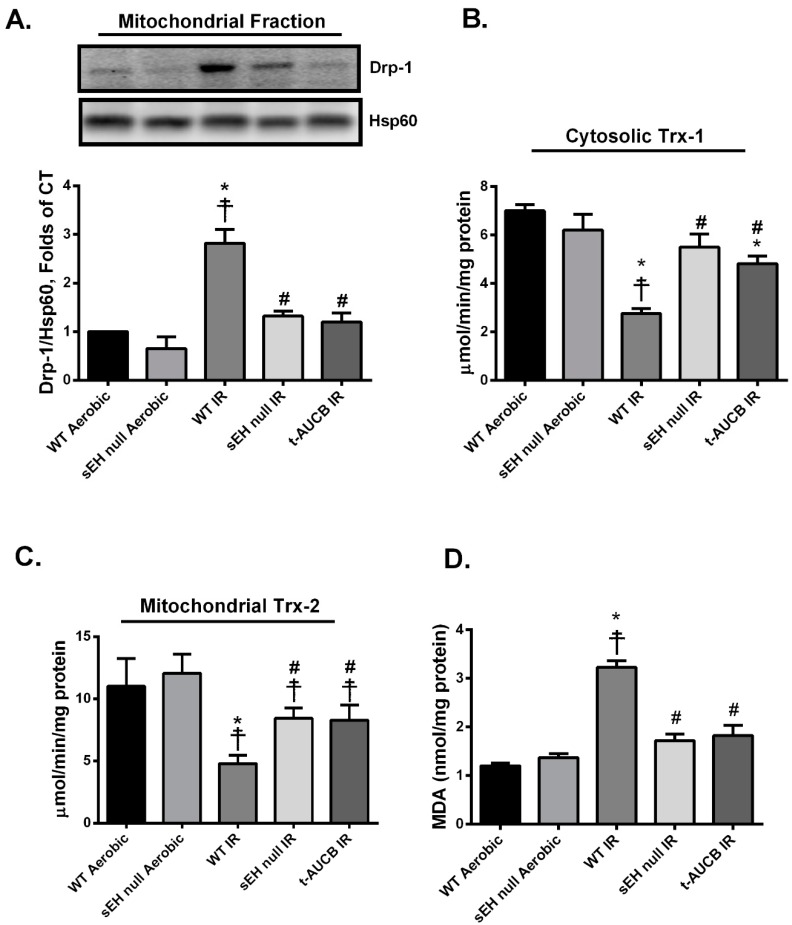
Deficiency of sEH preserved mitochondrial integrity and limited oxidative stress following ischemia/reperfusion (IR) injury. (**A**) Representative immunoblots and densiometric quantification of the expression of mitochondrial protein Drp-1 in mice hearts after 30 min ischemia and 40 min reperfusion. Protein expression was normalized to heat shock protein 60 (Hsp60) used as a loading control. Cardiac (**B**) cytosolic thioredoxin (Trx)-1 and (**C**) mitochondrial Trx-2 activities were assessed in hearts following 30 min ischemia and 40 min reperfusion. Trx activity was assessed using the insulin disulfide reduction assay. In this assay, by using an excess of NADPH and thioredoxin reductase (TrxR) enzyme, Trx activity was measured via the oxidation of NADPH and the generation of free SH groups in reduced insulin by 5,5’-dithiobis-(2-nitrobenzoic acid) (DTNB) after stopping the reaction with guanidine–HCl. (**D**) Cardiac MDA levels assessed using a lipid peroxidation (MDA) colorimetric assay kit in mice hearts following 30 min ischemia and 40 min reperfusion. Values represent mean ± SEM, * *p* < 0.05 vs. WT Aerobic CT, ☨ *p* < 0.05 vs. sEH null Aerobic, # *p* < 0.05 vs. WT IR (*n* = 3–5 per group).

**Figure 4 ijms-20-03502-f004:**
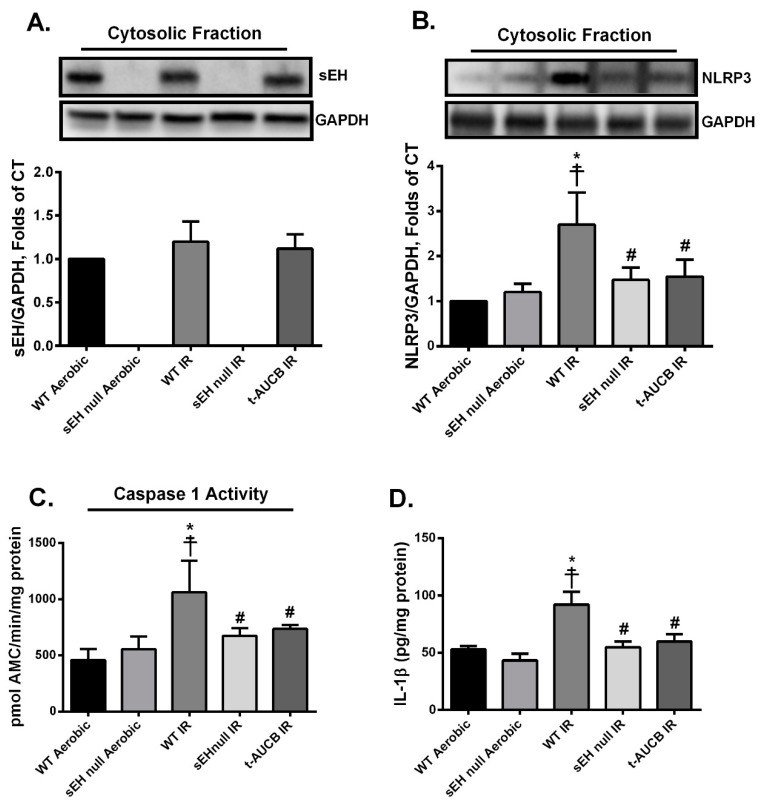
Genetic deletion of *Ephx2* or pharmacological inhibition of sEH inhibited IR-induced nucleotide-binding oligomerization domain-like receptor (NLR) family pyrin domain containing 3 (NLRP3) inflammasome formation and activation. Representative immunoblots and densiometric quantification of the expression of the cytosolic proteins (**A**) sEH and (**B**) NLRP3 in mice hearts after 30 min ischemia and 40 min reperfusion. All expressions were normalized to glyceraldehyde 3-phosphate dehydrogenase (GAPDH) loading control. (**C**) Cardiac caspase-1 enzymatic activity assessed in the cytosolic fraction following 30min ischemia and 40min reperfusion. The assay quantitated the fluorescence intensity resulting from the cleavage of the caspase-1 specific fluorogenic substrate Ac-YVAD-AMC by the cytosolic heart homogenates. (**D**) Cardiac IL-1β protein levels assessed by enzyme-linked immunosorbent assay (ELISA) in the cytosolic fraction following 30 min ischemia and 40 min reperfusion. Values represent mean ± SEM, * *p* < 0.05 vs. WT Aerobic CT, ☨ *p* < 0.05 vs. sEH null Aerobic, # *p* < 0.05 vs. WT IR (*n* = 3–5 per group).

**Figure 5 ijms-20-03502-f005:**
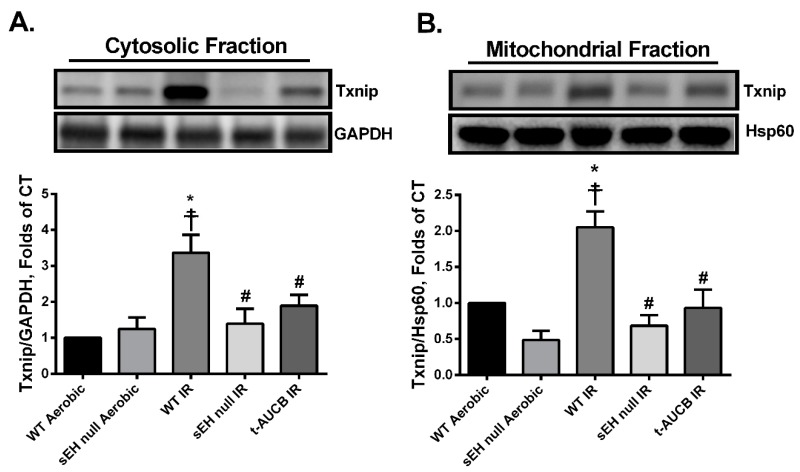
Genetic deletion of *Ephx2* or pharmacological inhibition of sEH prevented IR-induced activation and translocation of the pro-oxidant Txnip. Representative immunoblots and densiometric quantification of protein expression of (**A**) cytosolic Txnip and (**B**) mitochondrial Txnip in hearts after 30 min ischemia and 40 min reperfusion. Cytosolic protein expression was normalized to GAPDH while mitochondrial Txnip expression was normalized to Hsp60. Values represent mean ± SEM, * *p* < 0.05 vs. WT Aerobic CT, ☨ *p* < 0.05 vs. sEH null Aerobic, # *p* < 0.05 vs. WT IR (*n* = 3–4 per group).

**Figure 6 ijms-20-03502-f006:**
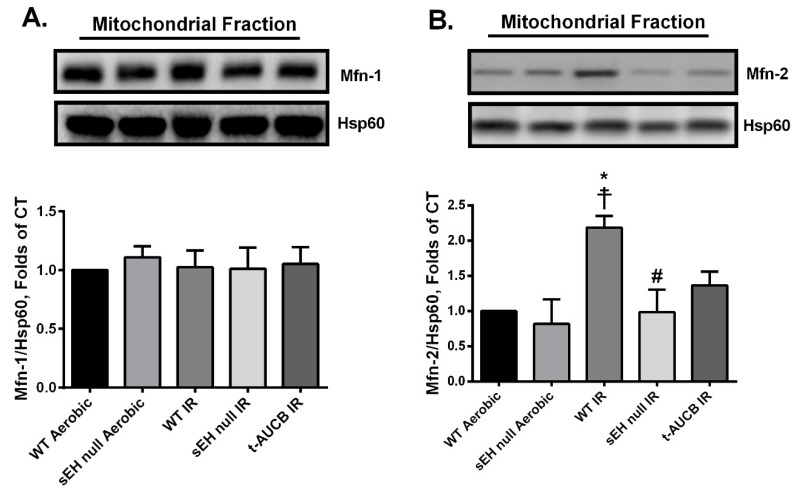
Effect of genetic deletion of *Ephx2* or pharmacological inhibition of sEH on the post-ischemic expression of mitochondrial mitofusins (Mfns). Representative immunoblot and densiometric quantification of the mitochondrial expression of (**A**) Mfn-1 and (**B**) Mfn-2 proteins in mice hearts following 30 min ischemia and 40 min reperfusion. Values represent mean ± SEM, * *p* < 0.05 vs. WT Aerobic CT, ☨ *p* < 0.05 vs. sEH null Aerobic, ^#^
*p* < 0.05 vs. WT IR (*n* = 3–4 per group).

**Figure 7 ijms-20-03502-f007:**
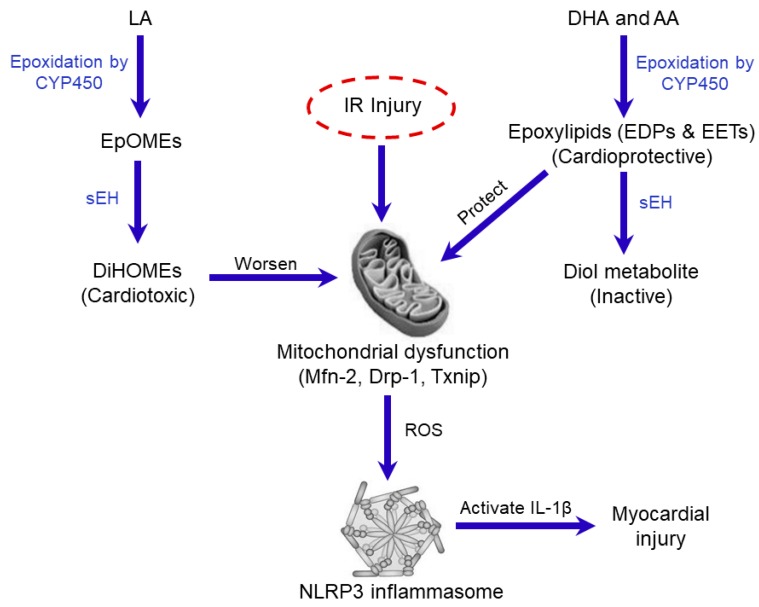
Schematic showing the potential roles of sEH in IR injury. Cellular injury in response to IR insult is associated with the release of polyunsaturated fatty acids (PUFAs) from the cell membrane which can be metabolized via epoxidation by cytochrome P450 (CYP450) isoenzymes to linoleic acid (LA) metabolites epoxyoctadecenoic acids (EpOMEs), which are further metabolized by sEH to dihydroxyoctadecenoic acid (DiHOMEs) with cardiotoxic effects targeting mitochondria resulting in injury. In contrast, sEH rapidly degrades the cardioprotective epoxylipids, epoxydocosapentaenoic acids (EDPs) and epoxyeicosatrienoic acids (EETs), generated from the CYP-mediated metabolism of the n-3 docosahexaenoic acid (DHA) and n-6 arachidonic acid (AA), respectively. These biologically active epoxy metabolites mediate many of the beneficial cardiovascular effects of the parent PUFAs by maintaining mitochondrial quality and reducing adverse inflammatory reactions. IR injury triggers the translocation of dynamin-related protein-1 (Drp-1) from the cytosol to the mitochondria initiating events that lead to the assembly and oligomerization of NLPR3 inflammasome. Active NLPR3 shuttles Txnip to the mitochondria aggravating mitochondrial damage as well the formation of the pro-inflammatory cytokine IL-1β triggering cardiomyocyte cell death.

**Table 1 ijms-20-03502-t001:** Mitochondrial respiration was measured in permeabilized cardiac fibers freshly isolated at the end of reperfusion.

Groups	Basal Respiration (nmol O_2_/min/mg)	ADP-Stimulated (nmol O_2_/min/mg)	Respiratory Control Ratio (RCR)
WT Aerobic	0.64 ± 0.12	3.36 ± 0.81	5.26 ± 0.49
sEH null Aerobic	0.58 ± 0.15	2.70 ± 0.71	4.90 ± 0.75
WT IR	0.75 ± 0.18	1.02 ± 0.21	1.43 ± 0.11 *☨
sEH null IR	0.97 ± 0.36	3.95 ± 1.37	4.03 ± 0.31 ^#^
*t*-AUCB IR	0.88 ± 0.18	2.92 ± 0.49	3.87 ± 0.59 ^#^

Oxygen consumption was assessed using a Clark electrode connected to an Oxygraph Plus recorder where malate and glutamate were used to stimulate basal respiration. Rates were presented as Respiratory Control Ratio (RCR), which is a ratio of adenosine diphosphate (ADP)-stimulated to basal respiration. Values represent mean ± SEM, * *p* < 0.05 vs. WT Aerobic CT, ☨ *p* < 0.05 vs. sEH null Aerobic, # *p* < 0.05 vs. WT IR (*n* = 4–8 per group).

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
