# Peer review of "Genetic Deletion or Pharmacological Inhibition of Soluble Epoxide Hydrolase Ameliorates Cardiac Ischemia/Reperfusion Injury by Attenuating NLRP3 Inflammasome Activation"

_ijms, 2019, doi:10.3390/ijms20143502_

Reviewer 1 Report

The manuscript determined the ability for sEH inhibition or sEH KO mice to protect the heart from ischemia reperfusion injury. Data provide significant evidence that sEH inhibition improves heart function in male and female mice through actions to improve mitochondrial function and decrease NLPR3 inflammasome. Overall, these data add to the mounting evidence that sEH inhibition protects the heart from ischemia reperfusion injury. 

Specific comment

Figure 6 details the fatty acid metabolites that could impact heart function through actions on the mitochondria, The levels or changes in these metabolites was not evaluated in this study. The authors need to acknowledge this limitation.

The food being fed to the animals is not provided. This is important because fatty acid content varies between diets.

Author Response

We would like to thank the reviewers for their time and effort reviewing our manuscript and greatly appreciate the opportunity to respond to the comments raised. We believe the revised manuscript which incorporates the reviewers’ suggestions has been significantly improved.

Response To Reviewer #1

1.   Figure 6 details the fatty acid metabolites that could impact heart function through actions on the mitochondria, the levels or changes in these metabolites was not evaluated in this study. The authors need to acknowledge this limitation.

We completely agree with the reviewer’s comment and believe analyses of the changes to fatty acid metabolites would markedly improve the manuscript.  While we were unable to assess these changes in the current study we are planning for our next study.  We have followed the reviewer’s suggestion and noted this limitation in the manuscript at the end of the discussion section. (Page 13, last paragraph of discussion, lines 389-394).

2.   The food being fed to the animals is not provided. This is important because fatty acid content varies between diets.

The reviewer raises an important point, which we missed. Our mice are fed a standard rodent chow diet from LabDiets (cat#5053).  It is composed of fat 11.3%, fiber 4.6%, protein 21% (w/w), more specifically it contains linoleic acid (2.12%), linolenic acid (0.27%), arachidonic acid (0.01%), omega-3 fatty acid (0.45%), total SFA (0.78%) and total MSFA (0.96%). We have added this information into the material and methods section 4.1, lines 399-404.

Reviewer 2 Report

The recommendation is to accept with minimal change.  This is a well written strong paper of clear basic and clinical significance.  The experiments are well designed and carried out.  Several things stand out as making this study particularly significant.  The workers used well documented in vivo studies of ischemia reperfusion damage and in steps took their work to the heart, then cell, the organelle level to show the role of inflammazome biology in the disease.  In addition, they studied both male and female animals separately and the lack of difference allowed them to combine their data.  Finally, they used both a genetic knock out and a chemical approach to test their hypothesis that the soluble epoxide hydrolase appears to make recovery from IR injury harder.   Minor comments are indicated below. Overall the reviewers hope that this exciting paper can be published quickly with minimal changes.

Line 16 and throughout.  Epoxygenase is a bad word and should not be use.  It is incorrectly used here.  I have used it but never again.  Replace this with ‘cytochromes P450’, or epoxidation by P450, or epoxy fatty acids.  The term epoxygenase suggests the P450 only makes epoxides.  That is the case for Bm epoxygenase (the Nobel prize this year) but it is wrong in the mammalian field.  There is no known epoxygenase to date in mammals.  It also suggests that only one or two enzymes are involved.  Hankinson has shown a variety of ‘xenobiotic metabolizing’ P450 can make epoxides of fatty acid.  They have other wording such as CYP-derived that is fine.

Remove both epoxygenases from figure 6 and the legend.

Line 85.    Use ital. for trans  all the way through

Line 86    Use ital. for t-  all the way through

Page 4.  Is the sEH message or protein increased.  Increasingly it appears both are indicators of inflammation as well as possible causes of inflammation.  This would be a nice addition.  Line 87: Should describe what LVDP is when it is first mentioned

Table 1: Having just the graph of Respiratory Control Ratio would be easier to understand if there is not a substantial deviation of the basal respiration.

Figure 2A: Western blot was normalized to HSP-60 level or GADPH. Ideally WBs should be normalized to multiple proteins and/or total protein level using Coomassie stain. Are these proteins not affected by IR and/or sEH deletion? It would be helpful to briefly explain how the activity of Trx-1 (and of caspase-1) were detected in the figure legend.

Why are the error bars for WT aerobic not visible in many graphs? Why is there a difference in standard deviation?

215   the NLRP3….

222 there is  mounting evidence that CYP

The discussion is nicely done.  Is there an evidence suggesting that DiHOME is increased in isolated hearts upon IR injury, or that there is enough EpOME/DiHOME produced naturally to have any effect? If the authors cite papers with quantitative data from the literature their argument would be stronger.

Since the authors seem to be having fun, they could address why we have the sEH and ways to regulate it, if it only seems to do bad things.  It is conserved from Pseudomonas through potatoes and people.  Why?

One could argue that the authors should have quantitative data on the effect if sEHI and sEH XO on lipids from normal and ischemic.  This reviewer’s view is that it would be nice – but this is an expensive and time consuming undertaking that would delay the paper.  So many studies have been done that doing it again here is not worth the cost in time or money. 

Author Response

We would like to thank the reviewers for their time and effort reviewing our manuscript and greatly appreciate the opportunity to respond to the comments raised. We believe the revised manuscript which incorporates the reviewers’ suggestions has been significantly improved.

Response To Reviewer #2

1.   Line 16 and throughout.  Epoxygenase is a bad word and should not be use.  It is incorrectly used here.  I have used it but never again.  Replace this with ‘cytochromes P450’, or epoxidation by P450, or epoxy fatty acids.  The term epoxygenase suggests the P450 only makes epoxides.  That is the case for Bm epoxygenase (the Nobel prize this year) but it is wrong in the mammalian field.  There is no known epoxygenase to date in mammals.  It also suggests that only one or two enzymes are involved.  Hankinson has shown a variety of ‘xenobiotic metabolizing’ P450 can make epoxides of fatty acid.  They have other wording such as CYP-derived that is fine.

We completely agree with the reviewer’s suggestion and comments.  We have traditionally used this terminology in the past and recognize its misuse, we would like to thank the reviewer for making note and we have followed the recommendation and replace the word “epoxygenase“  accordingly in the manuscript.

2.   Remove both epoxygenases from figure 6 and the legend.

This has been updated in the revised manuscript.

3.   Line 85. Use ital. for trans all the way through

We have amended this spelling as recommended.

4.   Line 86. Use ital. for t-  all the way through

We have amended this spelling as recommended.

 Page 4. Is the sEH message or protein increased.  Increasingly it appears both are indicators of inflammation as well as possible causes of inflammation. This would be a nice addition. 

We totally agree with the reviewer’s suggestion that linking alteration in sEH expression to the observed inflammatory response would be an interesting input to the current study. Accordingly, immunoblotting was performed to determine the changes in the levels of sEH protein in the WT mice in response to IR injury (Figure 4A) and it was found that there were no significant changes in the expression levels of sEH between the different WT groups in the setting of IR insults. However, not surprisingly in the short time course of the experiment, we did not assess either mRNA or catalytic activities. Previously, our group demonstrated that myocardial ischemia triggers an increase in sEH catalytic activity in WT hearts, which was inhibited by t-AUCB or deletion of Ephx2 gene [1,2]. These findings suggest alterations in the catalytic activity of sEH, and not its expression levels, is important for linking to inflammatory response in the setting of our isolated heart model of IR injury.  Unfortunately, we are unable to assess the changes to the metabolite profile (or sEH catalytic activity) in the study due to time and resource limitations. 

5.   Line 87: Should describe what LVDP is when it is first mentioned

We have clarified LVDP in the manuscript (Results section 2.1, line 88).

6.   Table 1: Having just the graph of Respiratory Control Ratio would be easier to understand if there is not a substantial deviation of the basal respiration.

We would like to thank the reviewer for this recommendation. We have followed the suggestion and included a bar chart representing the RCR values in the manuscript (Figure 2).

7.   Figure 2A: Western blot was normalized to HSP-60 level or GADPH. Ideally WBs should be normalized to multiple proteins and/or total protein level using Coomassie stain. Are these proteins not affected by IR and/or sEH deletion?

We recognize the reviewer’s suggestion, however, we, in addition to others, have showed that IR injury as well as deletion of the gene encoding for sEH (Ephx2) do not have any significant effect on the expression levels of the loading controls (i.e. HSP-60 and GAPDH) [1-3]. In that sense, we used these proteins as loading controls in the current study.

8.   It would be helpful to briefly explain how the activity of Trx-1 (and of caspase-1) were detected in the figure legend.

We completely agree with the reviewer’s suggestion and a brief description of the used methodology is added to the figure legends.

9.   Why are the error bars for WT aerobic not visible in many graphs? Why is there a difference in standard deviation?

We would like to thank the reviewer for making note of this point. We follow a standard protocol for densitometric quantification of the western blots where all the proteins are first normalized to the respective aerobic controls on the same blot and, next, the aerobic controls are normalized to together having the value of 1 with no standard deviation/error. Subsequently, all proteins are normalized to the loading controls.

10.  215   the NLRP3….

We have amended the sentence in the manuscript (Discussion, first paragraph, line 244).

11.  222 there is a mounting evidence that CYP

We have amended the sentence in the manuscript (Discussion, third paragraph, line 262).

12.  The discussion is nicely done.  Is there an evidence suggesting that DiHOME is increased in isolated hearts upon IR injury, or that there is enough EpOME/DiHOME produced naturally to have any effect? If the authors cite papers with quantitative data from the literature their argument would be stronger.

We would like to thank the reviewer for this recommendation. We have followed the suggestion and included more details in the discussion section (paragraph 3, line 270-271).

13.  Since the authors seem to be having fun, they could address why we have the sEH and ways to regulate it, if it only seems to do bad things.  It is conserved from Pseudomonas through potatoes and people.  Why?

The reviewer raises a great question and interesting point regarding the biology of sEH. While we do not have an answer nor the expertise, we have provided a brief new section in the discussion (second paragraph, Lines 249-259). 

14.  One could argue that the authors should have quantitative data on the effect if sEHi and sEH KO on lipids from normal and ischemic.  This reviewer’s view is that it would be nice – but this is an expensive and time-consuming undertaking that would delay the paper.  So many studies have been done that doing it again here is not worth the cost in time or money.

We would like to thank the reviewer for raising this issue as we completely agree with the opinion. However, we added this point as a limitation of the current study at the end of the discussion section. We are working toward getting these studies done in our next projects as they are important.

Reviewer 3 Report

Darwesh AM, et al. showed that sEH mediated metabolites participate in mitochondrial dysfunction under cardiac ischemia-reperfusion injury via activation of NLRP3 activation in the current study. These findings expand our understanding of the role of sEH in cardiac damage. And it may further be applied for cardiac protection in many clinical settings. There are several points need to be further addressed as follows:

Authors should provide the figures showing the area of infarction and area at risk for the hearts undergoing IR injury.

Mitochondrial morphology, including mitochondrial fragmentation, was found in a variety of heart diseases. However, authors didn't fully characterize the mitochondrial phenotype and function under inhibition of sEH. Drp1 translocation or Mfn expression alone cannot completely reflect the final consequences of sEH inhibition. Authors should provide the images to show the morphology/fragmentation/quantity of the mitochondria. Also, the examination on mitochondrial function is highly recommended as well.

Authors suggest that the inhibition of sEH attenuated NLRP3 inflammasome activation and therefore reduced cardiac IR injury by showing NLRP3 and IL-1β levels in Figure 3. However, the main consequence of NLRP3 mediated IL-1β release is to initiate a cascade of inflammation, eg. leukocyte recruitment, etc. Here, the authors used an isolated heart system to study the role of sEH, which may lack circulating immune cells. Authors should provide more evidence to support their hypothesis.

In the schematic model, DiHOMEs are critical intermediate metabolites, not only reflecting the activity of sEH, but also showing the protective vs detrimental pathway mediated by sEH. Therefore, it is very important to show the levels of these metabolites in the study.

Authors should detect the ROS levels in the hearts to show the link between mitochondrial dysfunction and NLRP3 activation.

Recommend using the abbreviation Ephx2, instead of sEH, when indicating the gene expression.

Author Response

We would like to thank the reviewers for their time and effort reviewing our manuscript and greatly appreciate the opportunity to respond to the comments raised. We believe the revised manuscript which incorporates the reviewers’ suggestions has been significantly improved.

Response To Reviewer #3

1.     Authors should provide the figures showing the area of infarction and area at risk for the hearts undergoing IR injury.

We recognize the reviewer’s suggestion and totally understand the importance of showing images for the area of infarction and area at risk for the hearts undergoing IR injury. However, the current study focused primarily on the assessment of the postischemic functional recovery of hearts subjected to IR injury. Unfortunately, we do not have enough sEH null mice to run the suggested new experiments. In addition, the time constraint to respond to the reviews was short (10 days) and the limited resources preclude our ability to run the recommended experiment.

2.     Mitochondrial morphology, including mitochondrial fragmentation, was found in a variety of heart diseases. However, authors didn't fully characterize the mitochondrial phenotype and function under inhibition of sEH. Drp1 translocation or Mfn expression alone cannot completely reflect the final consequences of sEH inhibition. Authors should provide the images to show the morphology/fragmentation/quantity of the mitochondria. Also, the examination on mitochondrial function is highly recommended as well.

We recognize the reviewer’s suggestion; however, our laboratory has previously showed that electron micrographs of both young WT and sEH null mice demonstrate similar mitochondrial ultrastructure and organization in the non-infarct region of the left ventricle in post-MI hearts (Figure below). However, in the infarct region, young WT mice displayed a complete loss of cellular organization, cristae and ultrastructure that was maintained in the young sEH null mice (Figure below) [1,5]. We included briefly this explanation in the discussion section (lines 354-356).

PLEASE SEE ATTACHED DOCUMENT.

Figure: Deletion of sEH encoding gene (Ephx2) protects the mitochondria from ischemic damage. Representative electron micrograph images of non-infarct and infarct regions of left ventricles from post-MI young WT and sEH null mice. (Magnification = 6000×) [5].

The current study estimated the effect of pharmacological inhibition of sEH or genetic deletion of Ephx2 on cardiac mitochondrial function and efficiency by analyzing mitochondrial respiration in situ using cardiac fibers isolated from both young males and females at the end of reperfusion. In line with these previously observed mitochondrial structural changes, the respiratory control ratio (RCR), a marker of mitochondrial efficiency, significantly decreased in WT mice of both sexes after IR injury. However, the RCR values were maintained in mice hearts where Ephx2 is genetically deleted or sEH is pharmacologically inhibited, suggesting more optimal mitochondrial oxidative respiration.

3.     Authors suggest that the inhibition of sEH attenuated NLRP3 inflammasome activation and therefore reduced cardiac IR injury by showing NLRP3 and IL-1β levels in Figure 3. However, the main consequence of NLRP3 mediated IL-1β release is to initiate a cascade of inflammation, eg. leukocyte recruitment, etc. Here, the authors used an isolated heart system to study the role of sEH, which may lack circulating immune cells. Authors should provide more evidence to support their hypothesis.

The reviewer raises an extremely important issue that we agree this is an unresolved question with our experimental model – ex vivo perfused heart.  While resident cardiac macrophages might have a role in the response, we agree the recruitment of leukocytes will have a minimal role in the response. As we perfuse our hearts for 40min of baseline, we believe any circulating leukocytes will most be removed and/or have a minimal effect in the observed response.  Our current view is the pro-inflammatory effect triggered by IL-1β is the result of direct detrimental effects on the myocardial itself.  We have expanded our discussion (lines 311-320) to better reflect these ideas.

4.     In the schematic model, DiHOMEs are critical intermediate metabolites, not only reflecting the activity of sEH, but also showing the protective vs detrimental pathway mediated by sEH. Therefore, it is very important to show the levels of these metabolites in the study.

We would like to thank the reviewer as we completely agree with the suggestion. However, due to time constraints and budgetary limitations we cannot assess these metabolites at this time.  We have added this point as a limitation of the current study at the end of the discussion section (last paragraph, lines 389-394).

5.     Authors should detect the ROS levels in the hearts to show the link between mitochondrial dysfunction and NLRP3 activation.

The reviewer raises an important issue regarding ROS levels and we agree assessment of ROS would be beneficial to provide further evidence of mitochondrial dysfunction.  However, we are limited in our ability to directly assess mitochondrial derived ROS levels as we only have frozen tissues at this time. Assessment of ROS levels is best preformed in fresh samples using techniques such as ‘spin-trapping’ as most of the ROS have short life span.  Again, due to time constraints and budgetary limitations we cannot assess these directly at this time.  However, we did assess changes in MDA levels, while not a direct determination of ROS, they are a marker of lipid peroxidation.  We have included the new data in the manuscript, Figure. 3D, which demonstrated inhibition or deletion of sEH attenuates IR injury induced increase in lipid peroxidation. Moreover, our data demonstrating that genetic deletion of Ephx2 or pharmacological inhibition of sEH limits post IR injury reduction in Trx-1 and -2 activities provides further evidences of reduced ROS in both the cytosolic and mitochondrial fractions.

6.     Recommend using the abbreviation Ephx2, instead of sEH, when indicating the gene expression.

We have amended this in the manuscript.

References

1.         Akhnokh, M.K.; Yang, F.H.; Samokhvalov, V.; Jamieson, K.L.; Cho, W.J.; Wagg, C.; Takawale, A.; Wang, X.; Lopaschuk, G.D.; Hammock, B.D., et al. Inhibition of Soluble Epoxide Hydrolase Limits Mitochondrial Damage and Preserves Function Following Ischemic Injury. Front Pharmacol 2016, 7, 133, doi:10.3389/fphar.2016.00133.

2.         Seubert, J.M.; Sinal, C.J.; Graves, J.; DeGraff, L.M.; Bradbury, J.A.; Lee, C.R.; Goralski, K.; Carey, M.A.; Luria, A.; Newman, J.W., et al. Role of soluble epoxide hydrolase in postischemic recovery of heart contractile function. Circ Res 2006, 99, 442-450, doi:10.1161/01.RES.0000237390.92932.37.

3.         Chiang, C.W.; Lee, H.T.; Tarng, D.C.; Kuo, K.L.; Cheng, L.C.; Lee, T.S. Genetic deletion of soluble epoxide hydrolase attenuates inflammation and fibrosis in experimental obstructive nephropathy. Mediators Inflamm 2015, 2015, 693260, doi:10.1155/2015/693260.

4.         Darwesh, A.M.; Jamieson, K.L.; Wang, C.; Samokhvalov, V.; Seubert, J.M. Cardioprotective effects of CYP-derived epoxy metabolites of docosahexaenoic acid involve limiting NLRP3 inflammasome activation (1). Can J Physiol Pharmacol 2019, 97, 544-556, doi:10.1139/cjpp-2018-0480.

5.         Jamieson, K.L.; Samokhvalov, V.; Akhnokh, M.K.; Lee, K.; Cho, W.J.; Takawale, A.; Wang, X.; Kassiri, Z.; Seubert, J.M. Genetic deletion of soluble epoxide hydrolase provides cardioprotective responses following myocardial infarction in aged mice. Prostaglandins Other Lipid Mediat 2017, 132, 47-58, doi:10.1016/j.prostaglandins.2017.01.001.

Round  2

Reviewer 3 Report

Authors provided additional evidence or further discussion to demonstrate the role of sEH in mitochondrial homeostasis and cardiac dysfunction under ischemia condition. Most of the comments have been addressed.